# PHINET V2: A MASK-FREE BRAIN-INSPIRED VISION REPRESENTATION LEARNING FROM VIDEO

## ABSTRACT

Recent advances in self-supervised learning (SSL) have revolutionized computer vision through innovative architectures and learning objectives, yet they have not fully leveraged insights from biological visual processing systems. Recently, a brain-inspired SSL model named PhiNet was proposed; it is based on a ResNet backbone and operates on static image inputs with strong augmentation. In this paper, we introduce PhiNet v2, a Transformer-based architecture that processes temporal visual input (that is, sequences of images) without relying on strong augmentation to learn robust visual representations, similar to human visual processing. Our learning objective is derived from variational inference. Through extensive experimentation, we demonstrate that PhiNet v2 achieves competitive performance compared to state-of-the-art vision foundation including RSP and CropMAE, while maintaining the ability to learn from sequential input without strong data augmentation. This work represents a step toward more biologically plausible computer vision systems that process visual information in a manner more aligned with human cognitive processes.

## 1 INTRODUCTION

Self-supervised learning (SSL) has emerged as a powerful paradigm in computer vision, advancing significantly through vision foundation models like SimCLR (Chen et al., 2020a), MoCo (Chen et al., 2020b), DINOs (Caron et al., 2021; Oquab et al., 2024), and Masked Autoencoders (MAE) (He et al., 2022). While these methods have achieved impressive results through architectural innovations and learning objectives, they have largely overlooked one of nature's most efficient learning systems; the human brain. Humans excel at learning visual representations through their ability to process continuous streams of visual information, suggesting that combining brain-inspired mechanisms with sequential processing can enhance machine learning approaches.

PhiNet (Ishikawa et al., 2025) (Figure 1b) marks a significant step toward brain-inspired learning. Drawing from biological circuitry in entorhinal cortex (EC), hippocampus and neocortex (Figure 1a), PhiNet combines the temporal prediction hypothesis (Chen et al., 2024) with Complementary Learning Systems (CLS) theory (McClelland et al., 1995). The model uses ResNet-based encoders, two predictors, and exponential moving average (EMA) modules, all trained end-to-end via backpropagation. This model, henceforth called PhiNet v1, has matched or exceeded the performance of leading ResNet-based methods including SimSiam (Chen and He, 2021), BYOL (Grill et al., 2020), and Barlow Twins (Zbontar et al., 2021), while maintaining robustness to weight decay choices.

While PhiNet v1 has shown promise in bridging machine learning with brain-inspired principles, it is not designed for sequential learning. In contrast, a truly comprehensive brain-inspired system should inherently support sequential processing, that is, the ability to learn from continuous streams of visual input, similar to how humans process visual information. In addition, PhiNet v1 has two other limitations. First, its ResNet-based architecture could be improved by incorporating a more powerful backbone (i.e., Transformers) to enhance representation learning. From a neuroscience perspective, recent studies suggest that Transformers closely resemble current models of the hippocampus (Whittington et al., 2022), making them a promising architectural choice for the next iteration of PhiNet. Second, its reliance, like SimSiam and BYOL, on strong data augmentation techniques limits its generalizability, particularly when learning from sequential input, where effective augmentations are harder to define.

We present PhiNet v2 (Figure 2b), a Transformer-based model that learns robust representations from sequential visual input. Our choice of input brings it closer to the work of Chen et al. (2024), which is an inspiration for PhiNet v1. We also replace PhiNet v1's ResNet-based encoder and predictor with Transformer-based modules, and incorporated an uncertainty model to capture the stochasticity in visual input (Jang et al., 2024). By careful architectural design incorporating these changes, we achieve stable and effective training. Our extensive experiments on standard computer vision tasks show that PhiNet v2 outperforms recent strong baselines, including DINO (Caron et al., 2021), SiamMAE (Gupta et al., 2023), and RSP (Jang et al., 2024). Ablation studies confirm that each architectural component of PhiNet v2 is crucial for stable and robust training.

Our contributions are summarized as follows:

- We propose PhiNet v2, the first brain-inspired vision representation learning based on a Transformer architecture.
- PhiNet v2 is capable of learning from sequential visual input without relying on strong data augmentation and masking, which is required by most existing vision foundation models.
- We formulate a variational learning objective for PhiNet v2, bridging brain-inspired modeling with probabilistic learning.
- PhiNet v2 achieves competitive or superior performance compared to strong pretraining methods such as DINO (Caron et al., 2021), SiamMAE (Gupta et al., 2023), CropMAE (Eymaël et al., 2024), and RSP (Jang et al., 2024).

## 2 RELATED WORK

In this section, we review the evolution of self-supervised learning methods in computer vision, from contrastive learning approaches to transformer-based architectures, leading to our the brain-inspired representation learning approaches.

Self-supervised learning (SSL) has emerged as a powerful paradigm for learning representations without explicit labels. A widely adopted approach is contrastive learning, where models are trained to pull together similar pairs (augmentations of the same image) while separating dissimilar pairs. SimCLR (Chen et al., 2020a) is a representative example, employing large batch sizes to mine hard negatives effectively. MoCo (He et al., 2020) introduces a momentum encoder and a queue-based dictionary, enabling the use of a dynamic memory bank to decouple batch size from the number of negative samples. MoCo v2 (Chen et al., 2020b) further improves this framework with stronger data augmentations and architectural changes.

To overcome the reliance on negative samples, several non-contrastive methods have been proposed. BYOL (Grill et al., 2020) and SimSiam (Chen and He, 2021) learn representations via asymmetric architectures, leveraging stop-gradient operations or momentum encoders to prevent representational collapse. Barlow Twins (Zbontar et al., 2021) introduces a redundancy reduction objective that encourages decorrelated feature representations without requiring negative samples. VICReg (Bardes et al., 2022) is an another regularization based method to avoid collapse. These methods are typically implemented using convolutional backbones such as ResNet.

With the advent of Vision Transformers (ViTs), transformer-based SSL methods have gained attention. DINO (Caron et al., 2021) adopts a self-distillation framework inspired by BYOL, while the Masked Autoendoder (MAE) (He et al., 2022) reconstructs masked image patches in an autoencoder-like setup. I-JEPA (Assran et al., 2023) generalizes these ideas by predicting contextual latent representations, bridging BYOL and MAE paradigms.

Recently, there have been many attempts to learn from sequences of images (i.e., videos). One of the early works is TimeSformer (Bertasius et al., 2021), a supervised learning method for video understanding. VideoMAE (Tong et al., 2022) introduced a self-supervised learning (SSL) approach using Transformer and masked autoencoding for video data. SiamMAE (Gupta et al., 2023) incorporates a Siamese architecture to enhance feature learning, achieving significant improvements over VideoMAE. CropMAE (Eymaël et al., 2024) further explores data augmentation strategies within the Siamese MAE framework. Meanwhile, RSP (Representation Learning with Stochastic Frame Prediction) (Jang et al., 2024) uses stochastic prediction in latent space to improve pixel prediction, showing improved performance over earlier MAE-based methods. These models generally rely on

Table 1: Comparison of SiamMAE, RSP, PhiNet v1, and PhiNet v2.

| Method | Backbone | Input | Target | Augmentation |
|---|---|---|---|---|
| SiamMAE (Gupta et al., 2023) | Transformer | Video | Pixel | Crop, Flip, Mask |
| RSP (Jang et al., 2024) | Transformer | Video | Pixel | Crop, Flip, Mask |
| PhiNet v1 (Ishikawa et al., 2025) | ResNet | Image | Latent | Crop, Flip, Blur, Jitter, ... |
| PhiNet v2 (Ours) | Transformer | Video | Latent | Crop, Flip |

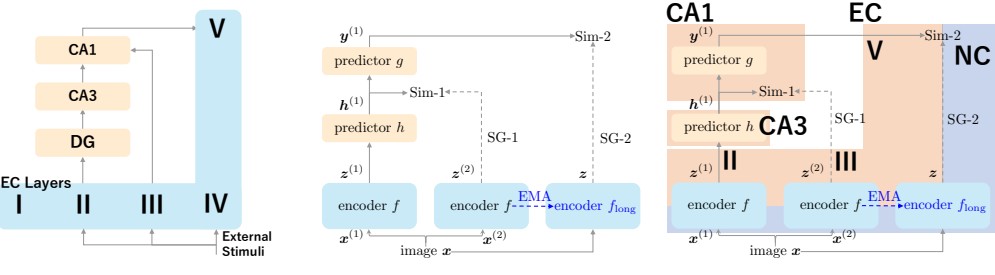

(a) The biological circuit.  (b) PhiNet v1 architecture.  (c) PhiNet v1 interpretation.

Figure 1: The biological circuit of the hippocampus and the biological circuit interpretation of PhiNet v1 (X-PhiNet in (Ishikawa et al., 2025)) and PhiNet v2. The hippocampus consists of CA1 and CA3 areas, and dentate gyrus (DG). As similar to (Chen et al., 2024), we assume that DG serves as a delay operator in Phinet models. EMA stands for exponential moving average; SG-1, SG-2, and SG-prior refer to the stop-gradient mechanism; and Sim-1 and Sim-2 are similarity/discrepancy functions.

pixel-level prediction tasks. In contrast, V-JEPA (Bardes et al., 2024) is a recent SSL method that performs masking and prediction in the representation space rather than the pixel space. Despite this, most existing video SSL approaches still focus on pixel-based prediction, and representation-space prediction remains relatively underexplored. Moreover, pixel prediction can be affected by noise, making it beneficial to learn representations without relying on pixel reconstruction.

PhiNet v1 (Ishikawa et al., 2025) introduces a biologically inspired architecture that shares structural similarities with SiamMAE but differs fundamentally in its learning objective. While SiamMAE and MAE focus on reconstructing images, PhiNet v1 learns by predicting latent representations. Additionally, PhiNet v1 employs a ResNet encoder instead of a transformer-based one. Table 1 provides a summary of key differences among SiamMAE, RSP, PhiNet v1, and our PhiNet v2 model.

## 3 BACKGROUND

Our proposed method is based heavily on neuroscience findings, and here we briefly review the key ideas behind this brain-inspired approach (Ishikawa et al., 2025).

### 3.1 NEUROSCIENCE FINDINGS

**Hippocampus and its biological circuit:** The hippocampus is a brain structure involved in memory. Figure 1a illustrates its biological circuitry. The hippocampus comprises several CA1 and CA3 regions, and the dentate gyrus (DG). External stimuli first enter the entorhinal cortex (EC), specifically layers II and III. From there, signals follow one of two pathways: EC layer II → DG → CA3 → CA1 or the direct route from EC layer III straight to CA1.

**Temporal Prediction Hypothesis: (Chen et al., 2024)** Recently proposed in computational neuroscience, the Temporal Prediction Hypothesis suggests that within the hippocampus, CA3 predicts upcoming inputs, while CA1 computes prediction error to update internal models. In this framework, CA3 processes current input and generates a prediction; CA1 then compares this prediction with the actual future input and uses the error signal for learning. DG may act as a temporal delay, aligning inputs with predictions. Recent experimental evidence supports this idea: neural recordings show that CA3 activity leads DG by around 2ms, consistent with a predictive role, while DG and CA1 are synchronized.

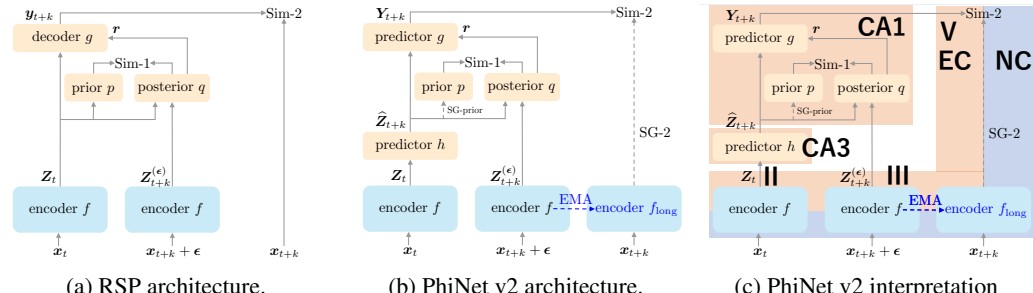

|     (a) RSP architecture.     |     (b) PhiNet v2 architecture.     |     (c) PhiNet v2 interpretation     |

Figure 2: The architectures of RSP (Jang et al., 2024), and PhiNet v2 are shown, illustrating that PhiNet v2 combines PhiNet v1 and RSP. The MAE module of RSP, which improves its performance, not shown, and it is not required by PhiNet v2.

**Complementary Learning Systems (CLS) (McClelland et al., 1995):** CLS theory posits that the brain uses two complementary learning systems: the hippocampus enables rapid, sparse learning of episodic information, while the neocortex gradually integrates experiences into distributed, overlapping representations. This dual-system organization allows new memories to be quickly stored by the hippocampus without overwriting existing knowledge, and later consolidated into neocortical structures through repeated interactions (e.g., replay processes).

## 3.2   PHINET V1

The PhiNet v1 model (also known as X-PhiNet in (Ishikawa et al., 2025)) is the first SSL method that implements both the temporal prediction hypothesis and the CLS theory. Figure 1b shows the architecture of the Phinet v1 model. The model consists of the Siamese encoder $f$, two predictors $h$ and $g$, and a slow encoder $f_{\text{long}}$, whose parameters are copied from $f$ using an exponential moving average (EMA). SG-1, SG-2, and SG-prior refer to stop-gradient mechanisms, and Sim-1 and Sim-2 denote similarity/discrepancy functions.

PhiNet v1 uses a ResNet backbone and is trained on images. To learn strong representations effectively, it applies heavy augmentations such as cropping, flipping, blurring, and color jitter.

Ishikawa et al. (2025) show, both empirically and theoretically, that the PhiNet v1 model is robust to the choice of the weight decay parameter. Furthermore, they empirically demonstrate that PhiNet v1 outperforms other existing representation learning methods in continual-learning scenarios.

However, PhiNet v1 has limitations. First, although it is designed based on the temporal prediction hypothesis, it does not utilize actual temporal information; instead, it simulates temporal differences using data augmentation and stop-gradient operations. Second, it relies on the ResNet architecture, and its performance with modern architectures—such as transformers—remains unclear.

## 4   PHINET V2

We propose PhiNet v2 based on the architectures of X-PhiNet (PhiNet v1) (Ishikawa et al., 2025) and RSP (Jang et al., 2024). To maintain easy and fair comparison to RSP, our code and parameters are based upon those of RSP.[1] Differences are shown in Figure 2 and outlined below.

**Compared with PhiNet v1**: First, PhiNet v1 operates on images with standard data augmentations, whereas PhiNet v2 is trained directly on video sequences, introducing additional temporal complexity. Second, while PhiNet v1 employs a multi-layer perceptron (MLP) as the predictor $g$, PhiNet v2 replaces it with transformer blocks to better capture spatio-temporal dependencies. Therefore, although the model structures of PhiNets v1 and v2 appear similar at a high level, achieving competitive performance with PhiNet v2 requires careful architectural and training considerations, as will be shown in our ablation study (Appendix C).

---

[1]https://github.com/huiwon-jang/RSP/

**Compared with RSP**: RSP (Jang et al., 2024) learns by predicting future pixel values while PhiNet v2 learns representations by aligning the predicted representation $\boldsymbol{Y}_{t+k}$ with the long-term representation $f_{\text{long}}(\boldsymbol{x}_{t+k})$. This shift from pixel-level prediction to representation-level alignment poses a significantly more challenging learning problem, such as overcoming representational collapse. Furthermore, while RSP relies on an additional masked autoencoder (MAE) module to enhance performance, PhiNet v2 achieves competitive results without such auxiliary components. Therefore, PhiNet v2 is a simpler yet effective alternative.

**Inputs**: In PhiNet v2, we aim to train the model using videos considered as related sequences of images. Let $\boldsymbol{x}_t^{(i)}$ and $\boldsymbol{x}_{t+k}^{(i)}$ denote two frames from a video, where $k$ is the frame sampling gap. The effective training set for our task is $\mathcal{D} = \{\{(\boldsymbol{x}_t^{(i)}, \boldsymbol{x}_{t+k}^{(i)})\}_{t=1}^{T_i}\}_{i=1}^{n}$, where $T_i$ is the number of frames in video $i$ and $n$ is the number of videos. For clarity, we will drop the superscript for the video.

The next few sections set the stage for PhiNet v2, culminating in section 4.4 where we concretize the model, in particular, using ViT as the encoder.

## 4.1 Brain-Inspired Interpretation

The PhiNet v1 model is inspired by the anatomical and functional architecture of the brain, particularly the interaction among the entorhinal cortex (EC), the hippocampus (CA3 and CA1) and the neocortex. PhiNet v2 is built upon the same: see Figures 1c and 2c for the brain-inspired interpretations of PhiNet v1 and PhiNet v2. Below, we follow PhiNet v1 and outline the components of PhiNet v2 with their biological counterparts.

**Entorhinal Cortex, EC-II/III/V, with Encoder** $f$: This module encodes sensory input (e.g., video frames) into low-dimensional representations, analogous to the role of the EC in relaying sensory information to the hippocampus. The encoded representations are forwarded to the hippocampal submodules (CA3 and CA1) for prediction. In PhiNet v2 where we use the ViT as the encoder $f$, the representations are in $\mathbb{R}^{d \times n_p}$, where $n_p$ is a number of tokens and $d$ is the dimension per token, and at time current time $t$ and future time $t + k$ they are

$$\boldsymbol{Z}_t = f(\boldsymbol{x}_t), \qquad\qquad \boldsymbol{Z}_{t+k}^{(\epsilon)} = f(\boldsymbol{x}_{t+k} + \boldsymbol{\epsilon}),$$

where noise $\boldsymbol{\epsilon}$ is added for the future, as in RSP. While the noise is not necessary for PhiNet v2, it can be motivated probabilistically (section 4.2) and it provides empirical improvement (Appendix C).

**CA3 Region, with Predictor** $h$: Based on the temporal prediction hypothesis (Chen et al., 2024), this module models the function of the CA3 region, which is hypothesized to generate predictions of future neural activity. In PhiNet v2, the predictor $h$ forecasts the representation of a future frame from that of the current frame:

$$\widehat{\boldsymbol{Z}}_{t+k} = h(\boldsymbol{Z}_t),$$

where $h : \mathbb{R}^{d \times n_p} \mapsto \mathbb{R}^{d \times n_p}$ is the predictor.

**CA1 Region, with Sim-1 Loss Function**: In alignment with hippocampal circuitry, CA1 receives both the predicted signal $\widehat{\boldsymbol{Z}}_{t+k}$ from CA3 and the actual future signal $\boldsymbol{Z}_{t+k}^{(\epsilon)}$ from the EC pathway. The network computes the discrepancy Sim-1 between these signals to enable rapid plasticity and fast learning through error-driven updates. In PhiNet v2, this discrepancy is modeled with a probability divergence between a distribution conditioned on $\widehat{\boldsymbol{Z}}_{t+k}$ and a distribution conditioned also on $\boldsymbol{Z}_{t+k}^{(\epsilon)}$ (Jang et al., 2024). Details are deferred to section 4.2, but one may refer to Figure 2b.

**CA1 Region, with Predictor** $g$: The CA1 region also outputs a signal to the V layer of EC for slow learning. While in PhiNet v1 this depends only on $\widehat{\boldsymbol{Z}}_{t+k}$, in PhiNet v2 inspired by RSP, it also relies on a low-dimensional random variable $\boldsymbol{r}$ dependent on actual future signals (see section 4.2):

$$\boldsymbol{Y}_{t+k} = g\left(\widehat{\boldsymbol{Z}}_{t+k}, \boldsymbol{r}\right),$$

where $g : \mathbb{R}^{d \times n_p} \times \text{Dom}(\boldsymbol{r}) \mapsto \mathbb{R}^{d \times n_p}$ is the predictor.

**Slow Learning Mechanism in Neocortex, with Slow Encoder** $f_{\text{long}}$ **and Sim-2 Loss Function**: This module mimics the slow-learning neocortical system, as postulated by the Complementary Learning Systems (CLS) theory. To implement this, Ishikawa et al. (2025) use an exponential moving average

(EMA) to model the slow dynamics of the neocortical encoder, called $f_{\text{long}}$ for long-term memory. A second objective, Sim-2, aligns representations from the hippocampal system and neocortex, supporting the feature prediction hypothesis and enabling long-term consistency in representations.

## 4.2 PROBABILISTIC DERIVATION OF PHINET V2

We provide a probabilistic derivation of PhiNet v2. For this purpose, we do not distinguish between $f$ and $f_{\text{long}}$, and simply use $f$. Distinction is made in section 4.3 when considering learning dynamics.

To learn the encoder $f$, a desideratum is to maximize the encoded data likelihood $\ell = p(f(\boldsymbol{x}_1), f(\boldsymbol{x}_2), \ldots, f(\boldsymbol{x}_T))$ for a video with $T$ frames. We implement this by maximizing a lower bound objective on $\ell$ with respect to the parameters of $f$. Our model and approximation choices are aimed towards a *simple objective for variational learning* of $f$ that is also brain-inspired (section 4.1).

Chain rule gives $\ell = \prod_{t=1}^{T} \ell_t$, where $\ell_t = p(f(\boldsymbol{x}_t)|f(\boldsymbol{x}_{t-1}), \ldots, f(\boldsymbol{x}_1))$. We make two modeling assumptions. First, we use a uniform mixture model for $\ell_t$ so that $\ell_t = \sum_{k=1}^{K_t} \ell_{t,k}/K_t$, where components $\ell_{t,k} = p(f(\boldsymbol{x}_t)|f(\boldsymbol{x}_{t-k}))$ are skipped Markov models. Second, to model video which is at once highly variable and highly structured (Chung et al., 2015), we include stochastic dependence on a latent variable $\boldsymbol{r}$, so that $\ell_{t,k} = \int p(f(\boldsymbol{x}_t)|f(\boldsymbol{x}_{t-k}), \boldsymbol{r}) \, p(\boldsymbol{r}|f(\boldsymbol{x}_{t-k})) \, \mathrm{d}\boldsymbol{r}$.

We begin with the classical evidence lower bound (ELBO) on $\ell_{t,k}$:

$$\log \ell_{t,k} \geq \int q(\boldsymbol{r}) \log p(f(\boldsymbol{x}_t)|f(\boldsymbol{x}_{t-k}), \boldsymbol{r}) \mathrm{d}\boldsymbol{r} - D_{\text{KL}}(q(\cdot)\|p(\cdot|f(\boldsymbol{x}_{t-k}))),$$

where the Kullback-Leibler (KL) divergence $D_{\text{KL}}$ is from the approximate posterior $q$ to the conditional prior $p$ of the latent variable $\boldsymbol{r}$. Then, one can show that

$$\log \ell \geq \frac{1}{K_t} \sum_{t=1}^{T} \sum_{k=1}^{K_t} \iint q(\boldsymbol{r}|\boldsymbol{\epsilon}) \log p(f(\boldsymbol{x}_t)|f(\boldsymbol{x}_{t-k}), \boldsymbol{r}) \mathrm{d}\boldsymbol{r} - D_{\text{KL}}(q(\cdot|\boldsymbol{\epsilon})\|p(\cdot|f(\boldsymbol{x}_{t-k})))q(\boldsymbol{\epsilon})\mathrm{d}\boldsymbol{\epsilon},$$

where $q(\boldsymbol{\epsilon})$ is a mixing distribution. The detailed derivation can be found in the appendix B.1. Thus far, the formulation is auto-regression by predicting from history. To complete the connection to the brain-inspired interpretation (section 4.1), particularly to the temporal prediction hypothesis in CA3 region, we shift the indices by $k$ to predict the future from current to get lower bound

$$\frac{1}{K_t} \sum_{t=1}^{T} \sum_{k=1}^{K_t} \iint q(\boldsymbol{r}|\boldsymbol{\epsilon}) \log p(f(\boldsymbol{x}_{t+k})|f(\boldsymbol{x}_t), \boldsymbol{r}) \mathrm{d}\boldsymbol{r} - D_{\text{KL}}(q(\cdot|\boldsymbol{\epsilon})\|p(\cdot|f(\boldsymbol{x}_t)))q(\boldsymbol{\epsilon})\mathrm{d}\boldsymbol{\epsilon}. \quad (1)$$

We now instantiate with the variables defined in section 4.1. The distributions of the latent variable $\boldsymbol{r}$ are functions of the conditioning variables: the prior is $p(\cdot|\widehat{\boldsymbol{Z}}_{t+k})$, and a component of the approximate posterior is $q(\cdot|\widehat{\boldsymbol{Z}}_{t+k}, \boldsymbol{Z}_{t+k}^{(\epsilon)})$ using amortized inference (Gershman and Goodman, 2014; Rezende et al., 2014). The combined KL divergences from the approximate posteriors to the priors is the Sim-1 loss in the CA1 region (Jang et al., 2024). The probability of $f(\boldsymbol{x}_{t+k})$ within $p(f(\boldsymbol{x}_{t+k})|f(\boldsymbol{x}_t), \boldsymbol{r})$ is conditioned on $\boldsymbol{Y}_{t+k}$, and it forms the basis of the Sim-2 loss.

Furthermore, we fix the mixing distribution $q(\boldsymbol{\epsilon})$ and do not optimise for it, that is, there is no dependence on the data. This retains biological plausibility without complicating the model significantly. While the objective (1) above and the loss (2) later are similar to known objectives such as that in RSP (Jang et al., 2024), we are not aware of any existing exposition in the manner presented above.

## 4.3 CONSIDERATIONS FOR LEARNING

We consider how $f$ can be successfully learned from random initialization. As a common challenge in SSL, it is easy to collapse to trivial solutions since the overall objective contains distances between (functions of) the different evaluations of $f$ in Sim-1 and Sim-2. Even without the collapse, we can fall into slow-learning regimes because the gradients for the different compute paths from $f$ can negate and lead to negligible effective gradient for learning $f$.

One approach is to bias the paths from encoder $f$ differently. For Sim-1 (KL divergence), we can attribute $\alpha \in [0, 1]$ to the prior $p$ and $1 - \alpha$ to the approximate posterior $q$ during learning; this is

known as KL balancing (Hafner et al., 2021). The effective bias for the EC-II encoder (leftmost encoder in Figure 2b) is higher than $\alpha$ because it also has a path to the posterior.

In addition to KL balancing, the two paths from the EC-II encoder to Sim-1 suggest that one of them could be suppressed during learning. To effect this, we stop direct gradient for the EC-II encoder via prior $p$ (SG-prior in Figure 2b) (Grill et al., 2020).

Another approach is to decouple the learning between the encoders. In PhiNet v2, we stop direct gradient updates to the neocortical encoder $f_{\text{long}}$ (SG-2 in Figure 2b), which gives the target representation $f(\boldsymbol{x}_{t+k})$ in the likelihood . Using the brain-inspired interpretation of PhiNet v2, the parameters of this encoder is updated indirectly via EMA (Grill et al., 2020): $\boldsymbol{\xi}_{\text{long}} \leftarrow \gamma\boldsymbol{\xi}_{\text{long}} + (1 - \gamma)\boldsymbol{\xi}$, where $\boldsymbol{\xi}$ and $\boldsymbol{\xi}_{\text{long}}$ are the parameters of $f$ and $f_{\text{long}}$, and $\gamma \in [0, 1]$ is the EMA decay factor. Therefore, $f_{\text{long}}$ is different from the EC encoders $f$ during learning, but they are the same at convergence.

## 4.4 OVERALL OBJECTIVE

We summarize the discussions above with an overall objective, after concretizing a few settings. First, the likelihood is isotropic multivariate normal with mean $\text{vec}(\boldsymbol{Y}_{t+k})$ and variance $\sigma^2 I$ for all but one token corresponding to the `[CLS]` in ViT (see section B.2), which is omitted and can be considered as having infinite variance. Second, we choose the latent variable $\boldsymbol{r}$ to be a random vector of $m$ $c$-categories variables (Hafner et al., 2021; Jang et al., 2024). Third, instead of summing over $k$ as written in section 4.2, we sample $k$ from $\mathcal{U}(k_{\min}, k_{\max})$. Omitting the normalizing constant for the multivariate normal likelihood, the loss is

$$\sum_{t=1}^{T}\left(\underbrace{\frac{1}{2\sigma^2}\|f_{\text{long}}(\boldsymbol{x}_{t+k}) - \boldsymbol{Y}_{t+k}\|_{\text{F}'}^2}_{\text{Sim-2}} + \underbrace{\sum_{i=1}^{m}\sum_{j=1}^{c}q_{tkij}\log q_{tkij}/p_{tkij}}_{\text{Sim-1}}\right), \qquad (2)$$

where $\|\cdot\|_{\text{F}'}$ is the Frobenius Norm excluding the `[CLS]`, $p_{tkij}$ is the prior probability of the $j$th category for $r_i$ when predicting the latent representation of $\boldsymbol{x}_{t+k}$ from that of $\boldsymbol{x}_t$; and $q_{tkij}$ is the corresponding approximate posterior probability. For the mixing distribution for the posterior, we use $\boldsymbol{\epsilon} \sim \mathcal{N}(0, \sigma_\epsilon^2 I)$. The variance in likelihood is $\sigma^2 = d(n_p - 1)\beta/2m$, where $\beta$ is a regularizing factor that plays the same role as the hyper-parameter typically weighing the $D_{\text{KL}}$ term (Denton and Fergus, 2018; Jang et al., 2024). Scaling $\sigma^2$ in this manner balances the output size of the encoder with the dimensions of the latent variable.

The loss is minimized with respect to the shared parameters $\boldsymbol{\xi}$ of the two encoders in the EC; the parameters of $g$ and $h$ in CA1 and CA2; and the parameters for the distributions $p$ and $q$. When computing gradients, we use KL balancing (section 4.3) and the straight-through estimator for $\boldsymbol{r}$ (Bengio et al., 2013; Jang et al., 2024).

**Symmetric loss**: We aim to promote bidirectional temporal consistency in the sense of both PhiNet v1 and SimSiam (Chen and He, 2021), and yet retain a valid probabilistic interpretation. To this aim, we apply the chain rule in section 4.2 in a reverse chronological order—$\ell = \prod_{t=1}^{T}\overleftarrow{\ell}_t$, where $\overleftarrow{\ell}_t = p(f(\boldsymbol{x}_t)|f(\boldsymbol{x}_{t+1}), \ldots, f(\boldsymbol{x}_T))$—and proceed as above to get the analogous reverse chronological loss. We sum the chronological and reverse chronological losses to obtain a symmetric loss.

## 5 EXPERIMENTS

We pretrain the encoder using the PhiNet v2 model and then use the encoder for downstream tasks. For pretraining, we use the Kinetics-400 (K400) dataset (Kay et al., 2017). We follow the preprocessing protocol in RSP (Jang et al., 2024). Pretraining is conducted on NVIDIA A100 and V100 GPUs.

For our method evaluation, we adopt the official evaluation scripts from prior works. Specifically, we use the RSP evaluation script[2] for the DAVIS dataset and the CropMAE evaluation script[3] for the

---

[2]`https://github.com/huiwon-jang/RSP/blob/master/eval_video_segmentation_davis.py`

[3]`https://github.com/alexandre-eymael/CropMAE/blob/main/downstreams/propagation/start.py`

Table 2: Results on video label propagation on three architectures (including ours) based on ViT-S/16. We report performances on video segmentation, video part segmentation, and pose tracking tasks from DAVIS, VIP, and JHMDB benchmarks, respectively. For all methods, we report the performance with the representations pre-trained on the Kinetics-400 dataset for 400 epochs. For CropMAE, we reevaluated based on the checkpoint provided in their github page. For fair comparison, we pretrained RSP with the same setup of PhiNet v2 with the author's of RSP code.

| Method | Davis | | | VIP | JHMDB | |
| --- | --- | --- | --- | --- | --- | --- |
| | $J\&F_m$ | $J_m$ | $F_m$ | mIoU | PCK@0.1 | PCK@0.2 |
| V-JEPA | 53.8 | 51.2 | 56.4 | 30.1 | 44.7 | 73.0 |
| CropMAE (Eymaël et al., 2024) | 57.0 | 54.8 | 59.3 | 33.0 | 43.4 | 71.8 |
| RSP (Jang et al., 2024) | 58.4 | 55.7 | 61.1 | 32.4 | 44.8 | 73.3 |
| PhiNet v2 | **60.1** | **57.2** | **63.0** | **33.1** | **45.0** | **73.6** |

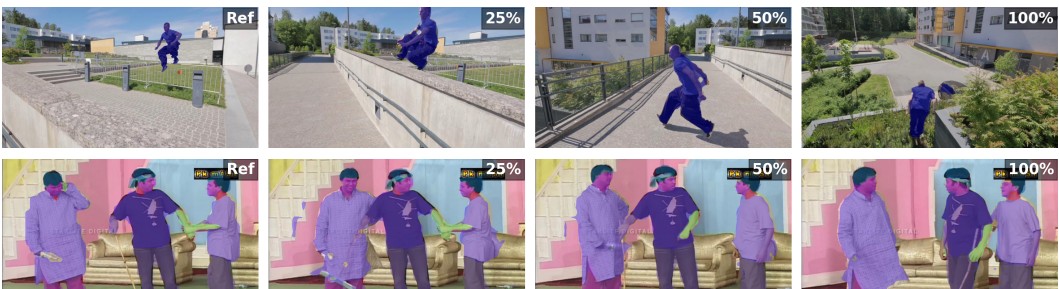

Figure 3: Qualitative results on the DAVIS and VIP datasets. "Ref" denotes the ground-truth mask in the first frame. "25%" and "75%" correspond to intermediate predictions, while "100%" shows the prediction on the final frame.

VIP and JHMDB datasets. We employ ViT-S/16 as the encoder with an embedding dimension of 384. The model is optimized using AdamW and trained for up to 400 epochs. Detailed hyperparameter settings for pretraining are listed in Table 10. We compare our PhiNet v2 model with RSP (Jang et al., 2024) and CropMAE (Eymaël et al., 2024), both which also use ViT-S/16.

## 5.1 RESULTS

Table 2 presents a comparison of the PhiNet v2 model with CropMAE and RSP. It shows that PhiNet v2 achieves performance comparable to both RSP and CropMAE. For a more comprehensive comparison with additional baselines, see Table 5.

RSP requires an additional MAE module to improve performance. The $J\&F_m$ score of RSP on the DAVIS dataset without the MAE module is reported by Jang et al. (2024) to be 57.7. In contrast, PhiNet v2 achieves 60.1 without relying on such a module. Since PhiNet v2 operates without an MAE module, it eliminates the need for tuning additional hyperparameters, making it more practical and easier to deploy.

Moreover, we evaluate the robustness of PhiNet v2 and RSP. One plausible hypothesis is that, by not relying on pixel-level reconstruction, our model becomes inherently more robust to anomalies or noisy frames. To validate this hypothesis, we conducted evaluations with Gaussian noise added to the input images and compared the performance between RSP and PhiNet v2. Table 6 shows that PhiNet v2 is more robust than RSP under noisy input for DAVIS experiments. For other cases, PhiNet v2 compares favorably with RSP.

Finally, Table 4 summarizes the ablation study and the proposed combination is important to get superior performance. The detailed analysis of the ablation study can be found in the Appendix C.

Figure 3 presents representative propagation results on the DAVIS dataset obtained using the PhiNet v2 model. The results demonstrate accurate and consistent propagation across frames.

Table 3: Results on video label propagation with different noise levels. In our experiment, we added Gaussian noise with standard deviations $\{0.0, 0.1, 0.2, 0.3\}$. We report performances on DAVIS, VIP, and JHMDB datasets. We report the performance with the representations pre-trained on the Kinetics-400 dataset for 400 epochs with the ViT small model. We set the regularization parameter $\beta = 0.01$.

| Dataset | Model | STD=0 | STD=0.1 | STD=0.2 | STD=0.3 |
|---|---|---|---|---|---|
| **DAVIS** | PhiNet v2 | 60.1 | 57.0 | 52.7 | 47.1 |
| | RSP | 58.4 | 55.7 | 49.6 | 44.2 |
| **VIP** | PhiNet v2 | 33.1 | 31.8 | 29.5 | 24.8 |
| | RSP | 32.4 | 31.1 | 28.7 | 24.4 |
| **JHMDB (PCK@0.1)** | PhiNet v2 | 45.0 | 45.1 | 44.3 | 41.8 |
| | RSP | 44.8 | 44.7 | 43.6 | 42.2 |
| **JHMDB (PCK@0.2)** | PhiNet v2 | 73.6 | 73.6 | 72.9 | 70.8 |
| | RSP | 73.3 | 73.0 | 71.9 | 70.2 |

Table 4: Results on video label propagation with symmetric and asymmetric loss functions and the existence of predictor $h$. We report performances on video segmentation using DAVIS benchmark, using the representations pre-trained on the Kinetics-400 dataset for 400 epochs with the ViT small model. We set the regularization parameter $\beta = 0.01$ and the batch size with 768. TF denotes the Transformer model in the table. *We report results at 96 epochs, which is the maximum number of epochs allowed for training. Note that the best model achieves a performance of 52.6 at 96 epochs, indicating a significant gap compared to other methods. †We replaced the transformer predictor with a linear predictor, applying a linear transformation only to $\widehat{Z}$ due to implementation constraints based on (Jang et al., 2024).

| PhiNet version | Symm. loss | $h$ | $g$ | $\epsilon$ | EMA | SG-prior | SG-post | $J\&F_m$ |
|---|---|---|---|---|---|---|---|---|
| v1 (Transformer) | – | – | – | – | – | – | – | 22.2 |
| v2 (Variants) | ✓ | ✓ | Linear$^\dagger$ | | ✓ | ✓ | | 26.8 |
| | ✓ | ✓ | TF | ✓ | | ✓ | | 34.9* |
| | ✓ | ✓ | TF | ✓ | ✓ | | ✓ | 55.8 |
| | | | TF | ✓ | ✓ | ✓ | | 58.0 |
| | | ✓ | TF | ✓ | ✓ | ✓ | | 58.7 |
| | ✓ | | TF | ✓ | ✓ | ✓ | | 58.9 |
| | ✓ | ✓ | TF | | ✓ | ✓ | | 59.3 |
| | ✓ | ✓ | TF | ✓ | ✓ | | | 59.7 |
| v2 (Proposed) | ✓ | ✓ | TF | ✓ | ✓ | ✓ | | 60.1 |

## 6 CONCLUSION

We have proposed PhiNet v2, a brain-inspired foundation model that learns robust video representations without relying on strong data augmentation. The model architecture builds upon the principles of the original PhiNet v1 (Ishikawa et al., 2025) and incorporates insights from the recent pixel-level prediction method RSP (Jang et al., 2024). Furthermore, we interpret PhiNet v2 through the lens of variational inference, providing a principled probabilistic framework that grounds the model in solid mathematical foundations. In contrast to RSP, which predicts raw pixel values and requires an auxiliary MAE module to boost performance, PhiNet v2 aligns latent representations and achieves comparable or superior results without additional components. This design choice not only simplifies the overall architecture but also enhances practicality by eliminating the need for extensive hyperparameter tuning. Experimental results across multiple benchmarks show that PhiNet v2 performs competitively with state-of-the-art methods, underscoring the promise of neuroscience-inspired approaches for scalable and generalizable video representation learning.

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

## A    THE USE OF LARGE LANGUAGE MODELS (LLMS)

We used a large language model (LLM) to assist with writing. Specifically, we drafted the sentences ourselves and then used the LLM to polish them. The LLM was not used to generate the ideas of this paper or to design the experiments. The authors take full responsibility for the content of the paper.

## B    FURTHER DETAILS OF PHINET V2

### B.1    PROBABILISTIC DERIVATION OF PHINET V2

We provide a probabilistic derivation of PhiNet v2. For this purpose, we do not distinguish between $f$ and $f_{\text{long}}$, and simply use $f$. Distinction is made in section 4.3 when considering learning dynamics.

To learn the encoder $f$, a desideratum is to maximize the encoded data likelihood $\ell = p(f(\boldsymbol{x}_1), f(\boldsymbol{x}_2), \dots, f(\boldsymbol{x}_T))$ for a video with $T$ frames. We implement this by maximizing a lower bound objective on $\ell$ with respect to the parameters of $f$. Our model and approximation choices are aimed towards a *simple objective for variational learning* of $f$ that is also brain-inspired (section 4.1).

Chain rule gives $\ell = \prod_{t=1}^{T} \ell_t$, where $\ell_t = p(f(\boldsymbol{x}_t)|f(\boldsymbol{x}_{t-1}), \dots, f(\boldsymbol{x}_1))$. We make two modeling assumptions. First, we use a uniform mixture model for $\ell_t$ so that $\ell_t = \sum_{k=1}^{K_t} \ell_{t,k}/K_t$, where components $\ell_{t,k} = p(f(\boldsymbol{x}_t)|f(\boldsymbol{x}_{t-k}))$ are skipped Markov models. Second, to model video which is at once highly variable and highly structured (Chung et al., 2015), we include stochastic dependence on a latent variable $\boldsymbol{r}$, so that $\ell_{t,k} = \int p(f(\boldsymbol{x}_t)|f(\boldsymbol{x}_{t-k}), \boldsymbol{r}) \, p(\boldsymbol{r}|f(\boldsymbol{x}_{t-k})) \, \mathrm{d}\boldsymbol{r}$.

We begin with the classical evidence lower bound (ELBO) on $\ell_{t,k}$:

$$\log \ell_{t,k} \geq \int q(\boldsymbol{r}) \log p(f(\boldsymbol{x}_t)|f(\boldsymbol{x}_{t-k}), \boldsymbol{r}) \mathrm{d}\boldsymbol{r} - D_{\text{KL}}(q(\cdot)\|p(\cdot|f(\boldsymbol{x}_{t-k}))),$$

where the Kullback-Leibler (KL) divergence $D_{\text{KL}}$ is from the approximate posterior $q$ to the conditional prior $p$ of the latent variable $\boldsymbol{r}$. Further application of Jensen's inequality gives

$$\log \ell_t \geq \frac{1}{K_t} \sum_{k=1}^{K_t} \int q(\boldsymbol{r}) \log p(f(\boldsymbol{x}_t)|f(\boldsymbol{x}_{t-k}), \boldsymbol{r}) \mathrm{d}\boldsymbol{r} - \frac{1}{K_t} \sum_{k=1}^{K_t} D_{\text{KL}}(q(\cdot)\|p(\cdot|f(\boldsymbol{x}_{t-k}))),$$

which can be interpreted as applying ELBO on the mixing proportions but using the prior for the posterior. This reflects our primary intent to learn the encoder via a simple objective rather than well-approximated posteriors. The lower bound on $\log \ell$ is the sum over $t$ of the above bound:

$$\log \ell \geq \frac{1}{K_t} \sum_{t=1}^{T} \sum_{k=1}^{K_t} \left( \int q(\boldsymbol{r}) \log p(f(\boldsymbol{x}_t)|f(\boldsymbol{x}_{t-k}), \boldsymbol{r}) \mathrm{d}\boldsymbol{r} - D_{\text{KL}}(q(\cdot)\|p(\cdot|f(\boldsymbol{x}_{t-k}))) \right).$$

We now enrich the approximate posterior $q(\boldsymbol{r})$ using the mixing construction $q(\boldsymbol{r}) = \int q(\boldsymbol{r}|\boldsymbol{\epsilon})q(\boldsymbol{\epsilon})\mathrm{d}\boldsymbol{\epsilon}$ (Jaakkola and Jordan, 1998). One can show that

$$\log \ell \geq \frac{1}{K_t} \sum_{t=1}^{T} \sum_{k=1}^{K_t} \iint q(\boldsymbol{r}|\boldsymbol{\epsilon}) \log p(f(\boldsymbol{x}_t)|f(\boldsymbol{x}_{t-k}), \boldsymbol{r}) \mathrm{d}\boldsymbol{r} - D_{\text{KL}}(q(\cdot|\boldsymbol{\epsilon})\|p(\cdot|f(\boldsymbol{x}_{t-k})))q(\boldsymbol{\epsilon})\mathrm{d}\boldsymbol{\epsilon}.$$

Thus far, the formulation is auto-regression by predicting from history. To complete the connection to the brain-inspired interpretation (section 4.1), particularly to the temporal prediction hypothesis in CA3 region, we shift the indices by $k$ to predict the future from current to get lower bound

$$\frac{1}{K_t} \sum_{t=1}^{T} \sum_{k=1}^{K_t} \iint q(\boldsymbol{r}|\boldsymbol{\epsilon}) \log p(f(\boldsymbol{x}_{t+k})|f(\boldsymbol{x}_t), \boldsymbol{r}) \mathrm{d}\boldsymbol{r} - D_{\text{KL}}(q(\cdot|\boldsymbol{\epsilon})\|p(\cdot|f(\boldsymbol{x}_t)))q(\boldsymbol{\epsilon})\mathrm{d}\boldsymbol{\epsilon}. \tag{3}$$

We now instantiate with the variables defined in section 4.1. The distributions of the latent variable $\boldsymbol{r}$ are functions of the conditioning variables: the prior is $p(\cdot|\widehat{\boldsymbol{Z}}_{t+k})$, and a component of the approximate posterior is $q(\cdot|\widehat{\boldsymbol{Z}}_{t+k}, \boldsymbol{Z}_{t+k}^{(\epsilon)})$ using amortized inference (Gershman and Goodman, 2014; Rezende

et al., 2014). The combined KL divergences from the approximate posteriors to the priors is the Sim-1 loss in the CA1 region (Jang et al., 2024). The probability of $f(\boldsymbol{x}_{t+k})$ within $p(f(\boldsymbol{x}_{t+k})|f(\boldsymbol{x}_t), \boldsymbol{r})$ is conditioned on $\boldsymbol{Y}_{t+k}$, and it forms the basis of the Sim-2 loss.

Furthermore, we fix the mixing distribution $q(\boldsymbol{\epsilon})$ and do not optimise for it, that is, there is no dependence on the data. This retains biological plausibility without complicating the model significantly.

While the objective (3) above and the loss (2) are similar to known objectives such as that in RSP (Jang et al., 2024), we are not aware of any existing exposition in the manner presented above.

### B.2   Implementation of PhiNet v2

The neural network setups are as follows:

**EC**: We adopt the Vision Transformer (ViT) (Dosovitskiy et al., 2021) as the backbone encoder $f$ for PhiNet v2. To process video input $\boldsymbol{x}$, we extract $n_p - 1$ non-overlapping image patches from each frame, add positional embeddings, and prepend one `[CLS]` token. The $n_p$ tokens are then processed by the ViT to obtained the representations.

**CA3**: We primarily employ a linear predictor for $h$: $\widehat{\boldsymbol{Z}}_{t+k} = \boldsymbol{W}_h \boldsymbol{Z}_t$, with $\boldsymbol{W}_h \in \mathbb{R}^{d \times d}$ the parameters of $h$. Although nonlinear variants of $h$ are possible, we have empirically found that linear predictors outperform more complex alternatives. This is likely because simpler predictors encourage the encoder $f$ to learn more expressive representations, while complex predictors may overfit and reduce the burden on the encoder.

**CA1**: Predictor $g$ is ViT-based with cross-attention where the keys and values for all transformer blocks are obtained from $\widehat{\boldsymbol{Z}}_{t+k}$ and $\boldsymbol{r}$, which are the arguments to $g$; and where the queries for the first transformer block are initialized with parameters to be learned but for the other layers are based on outputs from the previous transformer blocks (Jang et al., 2024; Chen et al., 2021).

For the distributions $p$ and $q$ on $\boldsymbol{r}$, we use single-hidden-layer neural networks with the ReLU activation. These neural networks use only the `[CLS]` tokens as inputs (Jang et al., 2024).

Table 5: Results on video label propagation. We report performances on video segmentation, video part segmentation, and pose tracking tasks from DAVIS, VIP, and JHMDB benchmarks, respectively. For all methods, we report the performance with the representations pre-trained on the Kinetics-400 dataset for 400 epochs.[†] refers to results reported in (Jang et al., 2024). We tried to compare the evaluation fairly by using the hyperparameter of RSP for VIP and JHMDB. However, since we are not using the same code for evaluation, the comparison of VIP and JHMDP might not be fair for some cases. Moreover, for RSP, we train their model with our environment using the author's Github code.

| Type | Method | Architecture | Davis | | | VIP | JHMDB | |
| --- | --- | --- | --- | --- | --- | --- | --- | --- |
| | | | $J\&F_m$ | $J_m$ | $F_m$ | mIoU | PCK@0.1 | PCK@0.2 |
| Autoencoder | MAE[†] | ViT-S/16 | 53.5 | 50.4 | 56.7 | 32.5 | 43.0 | 71.3 |
| | SiamMAE[†] | ViT-S/16 | 58.1 | 56.6 | 59.6 | 33.3 | 44.7 | 73.0 |
| | CropMAE | ViT-S/16 | 57.0 | 54.8 | 59.3 | 33.0 | 43.4 | 71.8 |
| | RSP | ViT-S/16 | 58.4 | 55.7 | 61.1 | 32.4 | 44.8 | 73.3 |
| Contrastive | SimCLR[†] | ViT-S/16 | 53.9 | 51.7 | 56.2 | 31.9 | 37.9 | 66.1 |
| | MoCo v3[†] | ViT-S/16 | 57.7 | 54.6 | 60.8 | 32.4 | 38.4 | 67.6 |
| Non-contrastive | Dino[†] | ViT-S/16 | 59.5 | 56.5 | 62.5 | 33.4 | 41.1 | 70.3 |
| | PhiNet v2 | ViT-S/16 | 60.1 | 57.2 | 63.0 | 33.1 | 45.0 | 73.6 |

## C   Ablation Study

Table 4 summarizes the ablation study. Below are details, with $J\&F_m$ scores in parenthesis.

**Algorithm 1** PhiNet v2 (Asymmetric version) Pytorch-like Pseudocode

```python
def forward(self, src_imgs, tgt_imgs):
    # Extract embeddings
    src_h = self.forward_encoder(src_imgs)
    tgt_p = self.forward_encoder(self.perturb(tgt_imgs))
    tgt_z = self.ema_model.forward_encoder(tgt_imgs)

    #CA3 output
    src_h_ca3_cls = self.ca3(src_h[:, 0])
    src_h_ca3 = self.ca3(src_h)

    # Posterior distribution from both images
    post_h = torch.cat([src_h_ca3_cls, tgt_p[:, 0]], -1)
    post_logits = self.to_posterior(post_h)
    post_dist = self.make_dist(post_logits)
    post_z = post_dist.rsample()

    # Prior distribution only from current images
    prior_h = src_h_ca3_cls
    prior_logits = self.to_prior(prior_h.detach())
    tgt_pred = self.forward_decoder_fut_latent(src_h_ca3, post_z)

    #Sim-1 (Hippocampal loss)
    kl_loss = self.kl_loss(post_logits, prior_logits)

    #Sim-2 (Neocortex loss)
    loss_post = mseloss(tgt_pred,tgt_z[:,1:,:].detach())

    loss = loss_post + self.kl_scale*kl_loss

    return loss
```

Table 6: Results on video label propagation with/without noise $\epsilon$. We report performances on video segmentation using DAVIS benchmark. We report the performance with the representations pre-trained on the Kinetics-400 dataset for 400 epochs with the ViT small model. We set the regularization parameter $\beta = 0.01$ and the batch size with 768.

| Noise level | 0 | 0.1 | 0.5 | 1.0 |
|---|---|---|---|---|
| $J\&F_m$ | 59.3 | 59.8 | 60.1 | 59.6 |

**PhiNet v1 with Transformer encoder (22.2)**  We investigate an extension of PhiNet v1 by replacing the ResNet encoder with a transformer. In this setting, we adopt the same input structure as PhiNet v2 and apply strong data augmentations to both $x_t$ and $x_{t+k}$. For the predictors $h$ and $g$, we use two-layer MLPs as from PhiNet v1 but without batch normalization because transformers already incorporate layer normalization. We also utilize an exponential moving average (EMA) module to mitigate collapse. As shown in Table 4, this variant performs poorly. This result highlights the necessity of a carefully designed architecture for transformer encoders.

**Exponential Moving Average (34.9)**  Although PhiNet v2 and RSP (Jang et al., 2024) share similar architectures, their representation learning dynamics differ. RSP avoids collapse by leveraging pixel-level prediction, whereas PhiNet v2 can suffer from representational collapse without additional stabilization. We empirically verified the importance of the exponential moving average (EMA) mechanism. As presented in Table 4, removing EMA results in collapsed representations and degraded performance, whereas including EMA stabilizes training and enables effective representation learning. This underscores EMA's essential role in the design of PhiNet v2.

**StopGradients (55.8, 59.7)**  The PhiNet v1 model uses StopGradient (SG) to stabilize training and introduce pseudo-temporal differences between input pairs. In PhiNet v2, since the input pairs

Table 7: Results on video label propagation with batch size {192, 384, 768, 1536}. We report performances on video segmentation using DAVIS benchmark. We report the performance with the representations pre-trained on the Kinetics-400 dataset for 400 epochs with the ViT small model. We set the regularization parameter $\beta = 0.01$.

| Batch size | 192 | 384 | 768 | 1536 |
|---|---|---|---|---|
| $J\&F_m$ | 59.2 | 59.9 | 60.1 | 58.9 |

Table 8: Results on video label propagation with difference regularization parameter $\beta$. We report performances on video segmentation using DAVIS benchmark. We report the performance with the representations pre-trained on the Kinetics-400 dataset for 400 epochs with the ViT small model. *We report results at 321 epochs, which is the maximum number of epochs allowed for training. Note that the model with $\beta = 0.01$ at the same epochs achieved 59.7.

| Parameter | $\beta = 0.001$ | $\beta = 0.01$ | $\beta = 0.03$ |
|---|---|---|---|
| $J\&F_m$ | 59.7 | 60.1 | 58.0* |

already contain temporal offsets, the necessity of SG-1 is uncertain. We conduct ablation to evaluate its impact. In this paper, we consider adding the SG operator to either prior side or posterior side.

As shown in Table 4, the absence of SG-prior leads to minor performance differences after 400 epochs (at 59.7). Interestingly, omitting SG-prior slightly improves early-epoch performance. Moreover, adding SG operator to the path of posterior, the performance significantly degraded (at 55.8).

**Symmetric Loss (58.0, 58.7)** We compare the performance of PhiNet v2 with symmetric versus asymmetric loss functions. Table 4 presents the results on the DAVIS benchmark using ViT-S pretrained for 400 epochs with a regularization parameter $\beta = 0.01$ and batch size of 768. The symmetric loss improves the final $J\&F_m$ score by 0.4 to 0.9, indicating its effectiveness in enhancing representation learning.

**Predictor (58.0, 58.9)** We also investigate the impact of the linear predictor module $h$ in CA3. Specifically, we have used a linear transformation $h(\boldsymbol{Z}) = \boldsymbol{W}_h \boldsymbol{Z}$, where $\boldsymbol{W}_h \in \mathbb{R}^{d \times d}$. As shown in Table 4, the predictor improves performance slightly. This is consistent with SimSiam (Chen and He, 2021), where the predictor plays a key role in avoiding collapse. In our model, the predictor contributes to learning better representations, especially when combined with the symmetric loss.

**Noise Term $\epsilon$ (59.3)** PhiNet v2 is inspired by both RSP (Jang et al., 2024) and PhiNet v1 (Ishikawa et al., 2025). While RSP introduces noise $\epsilon$ to prevent shortcut learning (e.g., copying pixels), we examine whether the a noise term is necessary for PhiNet v2—it improves the score by 0.8. Additional results (Table 6) suggest that PhiNet v2 does not benefit significantly from additional noise.

**Batch Size** We assess the robustness of PhiNet v2 to batch size by varying it across {192, 384, 768, 1536}. Table 7 shows that performance on the DAVIS benchmark remains stable across different batch sizes. Notably, PhiNet v2 achieves strong performance even with smaller batches, enabling training on limited hardware (e.g., V100 GPUs), which is a practical advantage.

**Regularization Parameter $\beta$** We investigate the effect of the regularization parameter $\beta$ by evaluating values in {0.001, 0.01, 0.03}. Table 9 shows the $J\&F_m$ scores on the DAVIS dataset under different $\beta$ values. Our results indicate that $\beta = 0.01$ yields a good trade-off, balancing stability and performance.

## D   LIMITATIONS

While PhiNet v2 demonstrates strong performance in more realistic settings and outperforms existing pixel based representation learning models, it has several limitations. First, although PhiNet v2 is inspired by the brain, it remains unclear to what extent it aligns with actual neural mechanisms in the human brain. Therefore, the current model should be regarded more as an engineering-driven approach rather than a biologically accurate one. Second, the encoder is based on the Vision Transformer architecture. However, the patch-based processing in Vision Transformers may not be the

Table 9: Results on video label propagation with different EMA parameters $\gamma$. We report performances on video segmentation using DAVIS benchmark. We report the performance with the representations pre-trained on the Kinetics-400 dataset for 400 epochs with the ViT small model.

| Parameter $\gamma$ | 0.95 | 0.98 | 0.99 | 0.999 |
|---|---|---|---|---|
| $J\&F_m$ | 49.5 | 60.0 | 60.1 | 57.8 |

Table 10: Hyperparameter details of pre-training.

| | Config | Value |
|---|---|---|
| Optimizer | Optimizer | AdamW |
| | Optimizer Momentum | $\beta_1 = 0.9, \beta_2 = 0.95$ |
| | Optimizer Weight Decay | 0.05 |
| | Learning Rate | $1.5 \times 10^{-4}$ |
| | Learning Rate Scheduler | Cosine decay |
| Training Schedule | Warmup Epochs | 40 |
| | Pre-train Epochs | 400 |
| Data | Repeated Sampling | 2 |
| | Effective Batch Size | 768 |
| | Frame Sampling Gap $[k_{\min}, k_{\max}]$ | [4, 48] |
| | Augmentation | hflip, crop [0.5, 1.0] |
| Model | The number of patches image patches $n_p - 1$ | 196 |
| | Output dimensions $d \times n_p$ of encoder $f$ | $384 \times 197$ (ViT-S/16 encoder) |
| | Discrete Latent Dimensions $m$ | 32 |
| | Discrete Latent Classes $c$ | 32 |
| | Mixing standard deviation $\sigma_\epsilon$ | 0.5 |
| | KL Balancing Ratio $\alpha$ | 0.8 |
| | EMA parameter $\gamma$ | 0.99 |
| | Frequency of EMA | Once per epoch |
| | Regularization parameter $\beta$ | 0.01 |

most faithful approximation of the human visual system. Third, the CA3 module is implemented as a linear encoder, whereas the biological CA3 region is known to have a recurrent structure. Exploring more biologically plausible alternatives, such as incorporating recurrence, is an important direction for future work.

# E  EXTENDED RELATED WORK

**Connection with Neuroscience**   Predictive coding was first proposed as a model of early visual processing in the retina, positing that sensory organs transmit only the unexpected ("error") components of their inputs to downstream areas (Srinivasan et al., 1982). This idea was later generalized to neocortical hierarchies, where each level predicts the activity of the level below and only prediction errors are propagated (Mumford, 1992; Rao and Ballard, 1999; Friston, 2005). By minimizing prediction errors across multiple scales of abstraction, the brain can efficiently encode complex sensory streams. Inspired by predictive coding, modern self-supervised and contrastive learning methods also leverage prediction-error–like objectives. Early approaches such as CPC (van den Oord et al., 2018) and data2vec (Henaff, 2020) learn representations by predicting masked or future latent features rather than reconstructing raw pixels. These works demonstrate that predictive objectives, even when formulated contrastively, can yield rich features for downstream tasks.

Chen et al. (2024) extended predictive coding to the hippocampal formation via the temporal prediction hypothesis, in which CA1 computes prediction errors between current inputs and CA3's generative model, and these errors are used to update CA3 representations. This aligns with the "predictive map" view of the hippocampus, in which place-cell and grid-cell networks encode a successor representation that predicts future states from the current one (Stachenfeld et al., 2017).

Table 11: Evaluation Hyperparameter details.

| Config | DAVIS | VIP | JHMDB |
|---|---|---|---|
| Top-k | 7 | 7 | 10 |
| Neighborhood size | 30 | 5 | 5 |
| Queue length | 30 | 3 | 30 |

Moreover, electrophysiological studies have shown that CA1 activity spikes when animals encounter unexpected deviations in a learned sequence, signaling mnemonic prediction errors that drive encoding of new episodes (Momennejad et al., 2018). Together, these findings reinforce the view that the hippocampus does not merely store episodic snapshots but actively forecasts upcoming inputs and uses the resulting error signals for rapid, one-shot learning. Several studies have explored leveraging hippocampal-inspired mechanisms for representation learning. For instance, Pham et al. (2021) introduced DualNet, a two-system architecture motivated by Complementary Learning Systems (CLS) theory (McClelland et al., 1995; Kumaran et al., 2016), whereby a slow, self-supervised pathway interacts with a fast, supervised pathway to balance stability and plasticity (Pham et al., 2021; 2023). DualNet demonstrated that combining self-supervised pretraining with supervised fine-tuning in a CLS-inspired loop can improve performance on a range of tasks.

Despite these advances, most existing models either focus on cortical predictive coding or apply CLS theory primarily in image-based contexts. In contrast, our PhiNet v2 integrates both hierarchical predictive coding and hippocampal temporal prediction within a unified, sequential learning framework that naturally processes video streams without relying on heavy data augmentation.

