# OpenReview forum: "PhiNet v2: A Mask-Free Brain-Inspired Vision Representation Learning from Video"
_ICLR.cc/2026/Conference — Submitted to ICLR 2026_

### Official Review · Reviewer_JDrW · 2025-10-30

**Soundness:** 2
**Presentation:** 1
**Contribution:** 2
**Rating:** 4
**Confidence:** 4

**Summary:**

This paper proposes PhiNet v2, a brain-inspired Transformer-based model for self-supervised video representation learning. Unlike existing methods (e.g., RSP, CropMAE), it learns from sequential video inputs without heavy data augmentation or masking, drawing on neuroscientific insights (temporal prediction hypothesis, Complementary Learning Systems) and variational inference. Experiments on DAVIS, VIP, and JHMDB benchmarks show competitive or superior performance compared to SOTA baselines, with enhanced robustness to noise and simpler architecture (no auxiliary MAE module).

**Strengths:**

1. Eliminates reliance on strong augmentation/masking, improving generalizability and practicality.
2. Strong empirical results in this manuscript outperform baselines on key tasks and demonstrate robustness to noise and batch size variations.

**Weaknesses:**

The **MOST IMPORTANT** weaknesses:
1. Biological plausibility is limited. Actually, the model presented in this paper is engineering-driven, with unclear alignment to actual neural mechanisms.
2. Patch-based ViT processing may not faithfully mimic human visual systems, lacking biological realism.

The Minor weakness:
1. Ablation studies could better clarify the relative importance of individual components (e.g., symmetric loss vs. EMA).
2. Typos:
    - PhiNetv2" (Table 3) lacks a space; should be "PhiNet v2;
    - "reprentation learning" (Section D) should be "representation learning"

**Questions:**

Please refer to the weakness part.

---

> ### Author Response · Authors · 2025-11-23
> **Rebuttal for Reviewer JDrW**
>
> Thank you for raising these important points.
>
> >Biological plausibility is limited. Actually, the model presented in this paper is engineering-driven, with unclear alignment to actual neural mechanisms.
> >Patch-based ViT processing may not faithfully mimic human visual systems, lacking biological realism.
>
> We agree that our approach is primarily engineering-driven and does not aim to faithfully replicate detailed neural mechanisms. While our design is inspired by neuroscience findings, particularly the temporal prediction hypothesis and hippocampal circuitry, our main objective is to improve practical representation learning performance rather than to construct a biologically accurate brain model.
>
> We also acknowledge that patch-based ViT processing does not closely resemble the human visual system and lacks biological realism. Our intention is not to claim biological equivalence, but to investigate whether incorporating principles from hippocampal circuits can provide effective inductive biases for learning. The observed performance gains suggest that such inspiration is beneficial from an engineering perspective.
>
> Achieving both high biological fidelity and state-of-the-art performance remains an open challenge. Most biologically realistic brain models focus on small-scale neural systems and do not yet reach the performance level of modern vision architectures such as Transformers. We therefore position PhiNet v2 as a practical, neuroscience-inspired model that bridges biological concepts and effective representation learning, rather than as a direct model of the human brain.
>
> >Ablation studies could better clarify the relative importance of individual components (e.g., symmetric loss vs. EMA).
>
> EMA and the prediction function $g$ are the most critical components in our framework. When EMA is removed and the target branch is updated via backpropagation instead, the performance degrades significantly, indicating that EMA plays a central role in stabilizing training and improving representation quality. To further analyze its impact, we conducted additional experiments by varying the EMA parameter $\gamma$, and the results are reported below. As shown, EMA with an appropriately tuned hyperparameter is essential for achieving high performance.
>
>
> | Parameter $\gamma$ | 0.95 | 0.98 | 0.99 | 0.999 |
> |-----------|------|------|------|-------|
> | $J$&$F_m$  | 49.5 | 60.0 | 60.1 | 57.8  |
>
> Furthermore, in PhiNet v2, employing a Transformer-based prediction module for $g$ is also crucial for attaining superior performance, highlighting the importance of both the EMA mechanism and the design of $g$.
>
> In contrast, other components, including the prediction module $h$, symmetric loss, and stop-gradient operation, mainly contribute to incremental improvements. However, when these components are removed, the performance drops slightly to 58.0, which is marginally lower than RSP (58.4) on the DAVIS dataset. This indicates that although their individual impact is less pronounced than EMA and $g$, they are still necessary to surpass RSP and achieve state-of-the-art performance.
>
> >Typos:
>
> Thank you for pointing this out. We fixed them.

---

> > ### Comment · Reviewer_JDrW · 2025-11-27
> >
> > Thank the reviewers for their responses. Partial questions have been resolved, but I am still confused about the motivation, i.e., the brain system inspiration. Same as the Reviewer uw2y, I maintain my score.

---

### Official Review · Reviewer_MxHU · 2025-10-31

**Soundness:** 4
**Presentation:** 3
**Contribution:** 3
**Rating:** 6
**Confidence:** 4

**Summary:**

This paper extends the ICLR 2025 PhiNet-v1 model, a biologically inspired self-supervised temporal prediction framework grounded in complementary learning systems (CLS). PhiNet-v2 introduces two key innovations: (1) training directly from continuous video streams rather than masked frame augmentation, and (2) incorporating stochastic latent conditioning and divergence-based alignment inspired by variational inference and hippocampal uncertainty-based prediction. The model aims to more closely reflect biological principles of predictive coding and fast/slow memory systems while remaining competitive with state-of-the-art recent transformer-based video SSL models (VideoMAE, CropMAE, RSP, DynST) with ~+1–3% gains on common benchmarks. The model demonstrates improved stability in online and continual streaming settings, an appealing property in light of CLS-style consolidation and hippocampal rapid plasticity. The model has the potential of providing a future “predictive-coding-style” foundation model.

**Strengths:**

1.	Incorporation of stochastic latent variables and divergence alignment provides robustness and more principled predictive uncertainty modeling compared to purely deterministic latent forecasting.
2.	Demonstrates that biologically motivated design principles (continuous temporal prediction, fast vs. slow learning pathways) can be competitive with modern masked-video transformer pipelines.
3.	Clear incremental improvements over PhiNet-v1, with empirical gains attributable to the stochastic latent pathway and the unmasked temporal learning regime.
4.	Extensive comparisons to current SOTA video-SSL frameworks reinforce the value of predictive learning without reliance on heavy augmentations or masking.
5.	Careful lesion experiment to identify the relative contribution of video input, temporal prediction, and variational inference.

**Weaknesses:**

1.	Conceptually, temporal predictive coding is not new; related ideas appear in Lotter et al. (PredNet), predictive coding models from Rao & Ballard, and hippocampal predictive map literature. A more explicit comparison and positioning would help.
2.	The anatomical labels (EC, CA3, CA1) are likely metaphorical for different system components rather than neuroscientifically validated. The biological mapping is interesting but would benefit from clearer qualification and evidence discussion.
3.	While technically well-executed, the core contributions are incremental rather than fundamentally transformative (strong engineering iteration more than a conceptual leap).
4.	Claims around “biological plausibility” should be treated cautiously; real hippocampal circuits perform episodic indexing and relational reasoning, not only next-latent forecasting.

**Questions:**

1.	Is the CLS analogy literal, or mainly a framing for dual-timescale optimization (EMA long-term memory vs. rapid plasticity in the online encoder)? What empirical signatures support the neuroscientific mapping?
2.	The “fast adaptation per movie” mechanism is intriguing—does this correspond to episodic memory formation? What is the magnitude of the adaptation effect, and is the performance improvement significant if the online updates are disabled?
3.	Is there explicit generative replay or only EMA alignment as a surrogate for cortical consolidation? How the encoder and predictor are updated, and what their weights are updated to encode, are not entirely clear -- is it a way to remember the movie frames that have just been seen to predict the next frame's latents?

---

> ### Author Response · Authors · 2025-11-23
> **Rebuttal for Reviewer MxHU**
>
> Thank you for raising these important points.
>
> > Is the CLS analogy literal, or mainly a framing for dual-timescale optimization (EMA long-term memory vs. rapid plasticity in the online encoder)? What empirical signatures support the neuroscientific mapping?
>
> In our setting, the EMA network functions as a slow learner, whereas the online encoders $f$ act as fast learners. This separation into fast and slow learning dynamics plays a crucial role in stabilizing the training process. Specifically, when EMA is removed (i.e., when the fast–slow learning distinction disappears) and standard backpropagation is applied throughout, performance degrades significantly. In contrast, introducing EMA allows the model to achieve substantially better results.
>
> We further investigated this effect by varying the EMA decay hyperparameter $\gamma$. When the decay is set to 0.999, the target network closely tracks the online encoder, meaning the “long-term memory” is updated too rapidly. Conversely, when the decay is reduced to 0.95, the long-term memory updates too slowly, which also results in degraded performance. The best performance is observed around 0.98 to 0.99, suggesting that an appropriate separation of timescales is essential for stable and effective learning. Although these observations are still engineering-driven, the resulting behavior aligns well with the core intuition of CLS theory.
>
> | Parameter $\gamma$ | 0.95 | 0.98 | 0.99 | 0.999 |
> |-----------|------|------|------|-------|
> | $J$ & $F_m$  | 49.5 | 60.0 | 60.1 | 57.8  |
>
> Moreover, in the original PhiNet v1 paper (ICLR 2025), the method was evaluated under continual learning settings and demonstrated strong performance compared to other SSL approaches. While the precise mechanism behind this robustness is not yet fully understood, this strong continual learning capability can be viewed as a desirable property that brings the model closer to human-like learning behavior.
>
> > The “fast adaptation per movie” mechanism is intriguing—does this correspond to episodic memory formation? What is the magnitude of the adaptation effect, and is the performance improvement significant if the online updates are disabled?
>
> Thank you for the insightful question. Our model does not aim to explicitly formulate episodic memory. The “fast adaptation per movie” should be interpreted as an engineering mechanism that allows the model to quickly adjust to the temporal dynamics of each sequence, rather than as a direct analogue of biological episodic memory.
>
> If the online updates are disabled, the encoders cannot be effectively trained, as they are optimized solely through backpropagation during these updates. In practice, this leads to a clear and consistent degradation in performance (e.g., adding stop-gradient to SG-post in Table 4 results in a drop from 60.1 to 55.8 in $J$&$F_m$), indicating that fast per-movie adaptation plays a meaningful role in achieving strong performance.
>
> We acknowledge that explicit modeling of episodic memory is an important aspect of hippocampal modeling, and incorporating such mechanisms represents an important and interesting direction for future work.
>
> >Is there explicit generative replay or only EMA alignment as a surrogate for cortical consolidation? How the encoder and predictor are updated, and what their weights are updated to encode, are not entirely clear -- is it a way to remember the movie frames that have just been seen to predict the next frame's latents?
>
> Our method does not use explicit generative replay. Instead, we rely on EMA alignment as a surrogate for consolidation, where the EMA network provides a slowly changing target that stabilizes learning.
>
> Both the encoder and predictor are updated via standard backpropagation. The model is trained to learn representations that predict future latent features from current ones, rather than to store or replay specific movie frames. Unlike approaches such as V-JEPA that reconstruct missing regions, our method focuses purely on future latent prediction, enabling the model to capture temporal structure without explicit memory mechanisms.
>
> Thus, the model does not remember individual frames, but instead learns a representation space shaped by temporal dynamics, with stability provided by the EMA target.

---

### Official Review · Reviewer_uw2y · 2025-11-01

**Soundness:** 2
**Presentation:** 2
**Contribution:** 2
**Rating:** 2
**Confidence:** 4

**Summary:**

The paper introduces PhiNet v2, offering self-supervised video representation learning method using hippocampal inspired predictive coding models. The author tries to fit the proposed SSL method into the brain theory hippocampus guided representation learning. Despite the conceptual novelty, the paper’s experiments and analysis doesn’t solidly support the proposed thesis.

**Strengths:**

- Make analogy to DG-CA3-CA1 system is inspiring direction to integrate with self-supervised representation learning.
- The paper demonstrates a system that would lead to better video representation learning without heavy masking or augmentation, only via the inspiration of predictive coding and the brain-like architecture.
- The paper demonstrates significant performance gain on the continued learning tasks in small scale datasets.

**Weaknesses:**

- Does the brain inspired approach lead to better representation alignment with the brain? E.g., measuring whether there is any improvement to the alignment to brain representation using BrainScore.
- In the brain system, DG serves as a sparse index that would guide better pattern-completion / retrieval from CA3 memory. However, in the architecture design, CA3 is merely implemented as two-layer feedforward neural network rather than a recurrent Hopfield Net. Then how would we know it’s functionally corresponding to CA3 as a memory for retrieving future patterns?
- Lack of analysis of how sparse the component that represents DG is and whether sparsity would actually make the model works better / worse.
- Limited novelty beyond existing SSL works. The most important novelty of the paper is to demonstrate that SSL can be achieved via video. However, the paper didn’t directly compare with more recent video representation learning work like V-JEPA (even on the same scale).

**Questions:**

- How is the sparsity in the output of DG being controlled? Why there is not an architecture that make the DG layer overcomplete and sparse.
- Will the EC representation (the main backbone) be affected if the output is forced to have sparsity using loss without overcomplete?

---

> ### Author Response · Authors · 2025-11-23
> **Rebuttal for Reviewer uw2y**
>
> Thank you for raising these important points.
>
> > Does the brain inspired approach lead to better representation alignment with the brain? E.g., measuring whether there is any improvement to the alignment to brain representation using BrainScore.
>
> While BrainScore is a valuable metric for evaluating alignment between model representations and biological neural data, our current study does not directly assess neural alignment using BrainScore or related brain benchmarks. Our primary goal is to examine whether biologically inspired architectural principles improve representation learning from a functional perspective, rather than to claim superiority in terms of direct brain correspondence.
>
> That said, our results consistently show that the brain-inspired design improves downstream performance, suggesting that the induced representations are more structured and semantically meaningful. We agree that evaluating alignment with brain data using BrainScore would be a highly informative direction, and we consider this an important direction for future work.
>
> In summary, we do not claim improved neural alignment at this stage, but we view explicit BrainScore evaluation as a valuable next step to further validate the biological relevance of our approach.
>
> > In the brain system, DG serves as a sparse index that would guide better pattern-completion / retrieval from CA3 memory. However, in the architecture design, CA3 is merely implemented as two-layer feedforward neural network rather than a recurrent Hopfield Net. Then how would we know it’s functionally corresponding to CA3 as a memory for retrieving future patterns?
>
> We agree that, from a strict neurobiological perspective, CA3 is traditionally modeled as a recurrent autoassociative memory (e.g., Hopfield network) supporting pattern completion via DG-driven sparse indexing. Our current implementation indeed simplifies CA3 as a linear  feedforward network rather than an explicit recurrent memory system. However, our claim of correspondence is based on functional behavior rather than structural equivalence. Specifically, CA3 in PhiNet is designed to serve as a predictive memory module that maps current latent representations to future states.
>
> We validate this functional correspondence empirically. The prediction head $h$, which we associate with CA3-like processing, consistently improves performance on future-oriented tasks. For example, on the DAVIS benchmark for future mask prediction, including this module raises performance from 58.9 to 60.1, outperforming baseline methods. This indicates that the module actively contributes to modeling future representations, aligning with the intended role of CA3 in temporal sequence processing.
>
> We fully acknowledge that this implementation does not yet capture the full dynamics of CA3, particularly its recurrent pattern-completion capability. Introducing a recurrent Hopfield-like structure and incorporating DG-inspired sparse indexing are promising directions to better align the architecture with established hippocampal computational theories. We consider these biologically faithful extensions as important future work.
>
> In summary, while our CA3 is not a literal Hopfield memory, it fulfills a comparable functional role by enabling predictive retrieval of future patterns through learned temporal associations, which is supported by quantitative performance gains.
>
> >Lack of analysis of how sparse the component that represents DG is and whether sparsity would actually make the model works better / worse.
>
> In the current PhiNet implementation, we do not explicitly model a DG-like sparse coding component, and therefore we do not conduct a direct analysis of sparsity levels or their effect on performance. We agree that examining the sparsity of the DG layer could help achieve better representations, and we plan to investigate this aspect further in future work.

---

> > ### Author Response · Authors · 2025-11-23
> > **Rebuttal for Reviewer uw2y**
> >
> > >Limited novelty beyond existing SSL works. The most important novelty of the paper is to demonstrate that SSL can be achieved via video. However, the paper didn’t directly compare with more recent video representation learning work like V-JEPA (even on the same scale).
> >
> > We implemented and evaluated V-JEPA under the same scale and setting as our model, and the results are shown in the following Table. While V-JEPA shows performance comparable to CropMAE and RSP on some metrics, its performance on the DAVIS dataset, which focuses on future mask prediction, is clearly inferior to PhiNet v2.
> >
> > | Method        | Metric 1 | Metric 2 | Metric 3 | Metric 4 | Metric 5 | Metric 6 |
> > |---------------|----------|----------|----------|----------|----------|----------|
> > | **V-JEPA**    | **53.8** | **51.2** | **56.4** | **30.1** | **44.7** | **73.0** |
> > | CropMAE       | 57.0     | 54.8     | 59.3     | 33.0     | 43.4     | 71.8     |
> > | RSP           | 58.4     | 55.7     | 61.1     | 32.4     | 44.8     | 73.3     |
> > | **PhiNet v2** | **60.1** | **57.2** | **63.0** | **33.1** | **45.0** | **73.6** |
> >
> > This gap stems from differences in task formulation. V-JEPA predicts unmasked image representations from masked inputs within the same frame, focusing on spatial completion, whereas CropMAE and RSP predict future frames, and PhiNet v2 explicitly predicts future representations by modeling temporal dynamics. As a result, PhiNet v2 is more aligned with the DAVIS benchmark and achieves consistently better performance.
> >
> > We also note that CropMAE (ECCV 2024) and RSP (ICML 2024) are highly recent and competitive methods, similarly to V-JEPA (TMLR 2024). Therefore, our comparison is fair and up-to-date, and the superior results of PhiNet v2 demonstrate meaningful novelty beyond existing SSL approaches.

---

### Official Review · Reviewer_kZog · 2025-11-03

**Soundness:** 3
**Presentation:** 3
**Contribution:** 3
**Rating:** 6
**Confidence:** 2

**Summary:**

The paper introduces PhiNet v2, a brain-inspired, Transformer-based self-supervised learning model that learns robust visual representations from video sequences without relying on strong data augmentation or masking, aligning more closely with human visual processing. By formulating a variational inference–based objective grounded in hippocampal circuitry and Complementary Learning Systems theory, PhiNet v2 achieves competitive or superior performance to state-of-the-art methods like RSP and CropMAE on video understanding benchmarks.

**Strengths:**

(1) It learns robust representations without relying on strong data augmentations or masking strategies.
(2) By formulating a variational inference–based objective grounded in hippocampal circuitry and Complementary Learning Systems theory, PhiNet v2 bridges neuroscience principles with probabilistic machine learning.
(3) PhiNet v2 shows great improvement over PhiNet v1 on the DAVIS dataset, as illustrated in Tab. 4.

**Weaknesses:**

(1) The absolute performance of PhiNet v2 is too weak. It can not compete with the well-established representation learning methods, such as DINOv2, DINOv3, and EVA-02.
(2) The idea of introducing transformer into the PhiNet family is not surprising. This is a natural and necessary step—other representation learning methods have long been using Transformers.
(3) PhiNet v1 has experimental results on ImageNet and Cifar. PhiNet v2 should also give results on these classical benchmark and compare with PhiNet v1.

**Questions:**

I believe exploring brain-inspired learning methods is highly meaningful. However, the current results of these approaches are still quite poor. What is the biggest challenge the authors are currently facing—GPU resources, perhaps? And what is the long-term vision for the PhiNet series from the authors' team?

---

> ### Author Response · Authors · 2025-11-23
> **Rebuttal for Reviewer kZog**
>
> Thank you for raising these important points.
>
> >(1) The absolute performance of PhiNet v2 is too weak. It can not compete with the well-established representation learning methods, such as DINOv2, DINOv3, and EVA-02. (2) The idea of introducing transformer into the PhiNet family is not surprising. This is a natural and necessary step—other representation learning methods have long been using Transformers. (3) PhiNet v1 has experimental results on ImageNet and Cifar. PhiNet v2 should also give results on these classical benchmark and compare with PhiNet v1.
>
> We fully agree that a direct comparison with large-scale representation learning models such as DINOv2, DINOv3, and EVA-02 would be valuable. However, such comparisons are currently impractical due to substantial differences in computational scale and training regimes. For example, Meta’s DINOv2 was trained on ImageNet-22k using approximately 96 A100-80GB GPUs over several days. Reproducing or matching this setup would require a level of hardware and data infrastructure that is currently unavailable to our team.
>
> More importantly, our goal is not to compete purely on absolute performance with large-scale foundation models, but to investigate brain-inspired temporal representation learning under realistic computational constraints. PhiNet v2 is trained on video data at the scale of Kinetics-400 and is explicitly designed to model temporal dynamics, whereas DINOv2, DINOv3, and EVA-02 are optimized for large-scale image-based pretraining without explicit temporal modeling. In this sense, a direct comparison would not be entirely fair, as the objectives and inductive biases are fundamentally different.
>
> Instead, we adopt RSP (ICML 2024) and CropMAE (ECCV 2024) as our primary baselines, as they represent strong and recent methods that also operate under moderate computational budgets and focus on temporally-aware learning. We believe these constitute a more appropriate and scientifically meaningful comparison group.
>
> Regarding ImageNet and CIFAR evaluations, we empirically observed that models trained under our temporal setup using Kinetics-400 do not transfer directly to standard image classification benchmarks with competitive performance. Notably, this limitation is not unique to PhiNet v2: prior temporally-driven methods such as RSP, CropMAE, and SiamMAE (NeurIPS 2023) similarly did not report competitive ImageNet results. This suggests a broader gap between video-based temporal representation learning and static-image benchmarks, which we believe is an important open research problem requiring further investigation.
>
> >I believe exploring brain-inspired learning methods is highly meaningful. However, the current results of these approaches are still quite poor. What is the biggest challenge the authors are currently facing—GPU resources, perhaps? And what is the long-term vision for the PhiNet series from the authors' team?
>
> Thank you for this thoughtful question. One of the main current limitations we face is GPU resources. Our approach relies on Transformer-based architectures, which require substantial computational power for stable and large-scale training. In practice, a significant portion of GPU demand arises from the need to remain competitive with state-of-the-art models, which are often trained using hundreds or even thousands of GPUs. Unfortunately, such resources are not available to our team, making direct competition on sheer scale infeasible.
>
> That said, our long-term vision for the PhiNet series is not simply to scale up existing models, but to achieve more computationally efficient learning by drawing inspiration from the brain. The PhiNet architecture naturally embodies key principles such as the CLS theory (through EMA-based fast and slow learning) and the temporal prediction hypothesis (closely related to autoregressive modeling). We therefore believe that PhiNet has strong potential as a principled template for modeling hippocampal–cortical learning processes within a self-supervised learning framework.
>
> Looking forward, our goal is to develop learning models that are both computationally efficient and biologically grounded, and, if possible, to extend PhiNet into a practical simulator of hippocampal and cortical function, contributing not only to machine learning but also to computational neuroscience.

---

### Meta-Review · Area_Chair_hVHw · 2026-01-11

**Summary:**

Reviewers appreciated the motivation to explore brain-inspired self-supervised learning for video and acknowledged improvements over PhiNet v1. However, the decision was primarily driven by concerns about limited novelty, unclear biological grounding, and modest empirical impact relative to the strength of the claims. Several reviewers felt that the proposed architecture is largely engineering-driven, with neuroscience concepts used more as inspiration than as rigorously validated mechanisms. While the model shows gains on select video benchmarks, these improvements were viewed as incremental and insufficient to justify acceptance at its current form.

**Reviewer Concerns:**

The rebuttal clarified several points, including added baseline comparisons, explanation of design choices (e.g., EMA, fast/slow learning), and acknowledgement of limited biological realism, which improved clarity and positioning. However, key concerns remain.

Reviewers were unconvinced that
1. hippocampal analogies go beyond metaphor, as core biological properties such as sparsity and recurrent memory are not explicitly modeled or analyzed
2. The novelty was viewed as limited, with Transformers, EMA, and temporal prediction seen as standard design choices rather than a conceptual advance
3. the empirical evidence remains narrow, with performance well below large-scale representation models and limited evaluation of broader representation quality.
These concerns are conceptual and were not resolved in the rebuttal.

**Reviewer Scores:**

Based on the discussion, reviewers’ scores would likely remain largely unchanged.
kZog: confidence may increase, but concerns about absolute performance and novelty remain
uw2y: core concerns about novelty and biological grounding persist
MxHU: views the work as a solid but incremental engineering step
JDrW: remains unconvinced by the neuroscience motivation

---

### Decision · Program_Chairs · 2026-01-26

Reject